# Low Doses of Celecoxib Might Promote Phenotype Switching in Cutaneous Melanoma Treated with Dabrafenib—Preliminary Study

**DOI:** 10.3390/jcm11154560

**Published:** 2022-08-04

**Authors:** Diana Valentina Tudor, Adrian Florea, Mihai Cenariu, Diana Elena Olteanu, Marius Farcaș, Andreea Hopârtean, Simona Valeria Clichici, Gabriela Adriana Filip

**Affiliations:** 1Department of Physiology, Faculty of Medicine, “Iuliu Hațieganu” University of Medicine and Pharmacy, 400012 Cluj-Napoca, Romania; 2Department of Cell and Molecular Biology, Faculty of Medicine, “Iuliu Hațieganu” University of Medicine and Pharmacy, 400349 Cluj-Napoca, Romania; 3Department of Animal Reproduction and Reproductive Pathology, University of Agricultural Sciences and Veterinary Medicine, 400372 Cluj-Napoca, Romania

**Keywords:** melanoma, celecoxib, dabrafenib, MITF, AXL

## Abstract

Background: Cutaneous melanoma is a heterogeneous tumor with a rapidly switching molecular and cellular phenotype. The invasive phenotype switching characterized by MITF^low^/AXL^high^ predicts early resistance to multiple targeted drugs in melanoma. Celecoxib proved to be a valuable adjuvant in cutaneous melanoma in preclinical studies. Our in vitro study evaluated for the first time whether celecoxib could prevent phenotype switching in two human melanoma cell lines treated with dabrafenib. Methods: All in vitro experiments were carried out on *BRAF-V600E*-positive A375 and SK-MEL-28 human melanoma cell lines, and subjected to a celecoxib and dabrafenib drug combination for 72 h. Melanoma cells were already in the MITF^low^/AXL^high^ end of the spectrum. Of main interest was the evaluation of the key proteins expressed in phenotype switching (TGF-β, MITF, AXL, YAP, TAZ), as well as cell death mechanisms correlated with oxidative stress production. Results: Celecoxib significantly enhanced the apoptotic effect of dabrafenib in each melanoma cell line compared to the dabrafenib group (*p* < 0.0001). Even though celecoxib promoted low MITF expression, this was correlated with high receptor tyrosine kinase AXL levels in A375 and SK-MEL-28 cell lines (*p* < 0.0001), a positive marker for the phenotype switch to an invasive state. Conclusion: This preliminary study highlighted that celecoxib might promote MITF^low^/AXL^high^ expression in cutaneous melanoma treated with dabrafenib, facilitating phenotype switching in vitro. Our results need further confirmation, as this finding could represent an important limitation of celecoxib as an antineoplastic drug.

## 1. Introduction

Cutaneous melanoma is one of the most heterogenic tumors, accounting for ~75% of skin-cancer-related deaths worldwide. More than half are *BRAF*-positive melanomas, 90% of which have *BRAF^V^*^600E^ substitutions [1]. Understanding melanoma biology is essential in order to overcome therapeutic resistance using patient-centered targeted treatment options [2]. Recent efforts have focused on piecing together melanoma resistance mechanisms in order to extend targeted therapeutic options and prolong the treatment response of advanced stages for melanoma patients.

Several forms of resistance to targeted therapies are described. These include MAPK pathway reactivation or substitutive pathways activation, extracellular matrix remodeling, autophagy upregulation, and the selection of a resistant tumor cell subpopulation. Another strategy that allows melanoma cells to bypass BRAF inhibitors (BRAFi) treatment is not genetic, but implies phenotype switching. This facilitates the rewiring of the transcriptional program with the further accommodation of melanoma cells to stressful conditions. It also includes a differentiation/de-differentiation switch linked to the expression of the microphthalmia transcription factor (MITF) and receptor tyrosine kinase AXL, as well as changes in proliferation rates. MITF is essential for melanocyte development, it regulates the expression of pigment-producing enzymes and also plays a central role in melanoma: it induces the escape from innate immunity, reprograms focal adhesion and the extracellular matrix, and regulates starvation-induced autophagy [3,4,5]. *BRAF* regulates MITF expression via PAX3 and BRN2 transcription factors, which act as a rheostat [6]. In addition, MITF and BRN2 together impact AXL expression. In aggressive melanoma tumors, AXL and MITF levels inversely correlate, showing MITF^low^/AXL^high^ expression characterized by a slow proliferating state, invasion, and metastasis formation. This phenomenon was described in melanoma tumors treated with cisplatin or BRAF/MEK inhibitors [7,8,9]. A similar pattern was recently depicted as a cause of resistance to immune checkpoint inhibitors as well. It seems that the loss of the MITF favors the upregulation of integrin beta-like protein 1 (ITGBL1) and decreases the cytotoxicity of immune cells, promoting immune escape. Thus, melanoma cells secrete negative immunomodulatory agents whose release is increased upon MITF silencing [10].

Cyclooxygenase-2 (COX-2) overexpression is correlated with melanoma initiation and progression, proving a negative prognostic marker [11]. Celecoxib, a selective COX-2 inhibitor, gained a lot of interest for this reason. Celecoxib might be a good candidate for drug repurposing in cancer because of its capacity to inhibit angiogenesis, cell invasion, and evasion from immune control, in order to reduce the negative response to BRAF inhibitors, correlated with a lower risk for cardiovascular events compared to other non-steroidal anti-inflammatory drugs [11,12,13]. Even at low doses, celecoxib demonstrated an adjuvant potential in melanoma co-culture after preventing phosphatidylinositol 3-kinase/protein kinase B (PI3K/AKT) activation, reducing melanogenesis, and promoting apoptosis. Even so, its exact antineoplastic mechanism of action is still under investigation and postpones a clear clinical validation. In a previous study, we showed that celecoxib and trametinib potentiated each other’s cytotoxic capabilities, proving a real antineoplastic synergism in an experimental melanoma model. Given the clinical use of dabrafenib and trametinib association in late stages of melanoma, it would be interesting to evaluate if this synergism is still preserved between celecoxib and dabrafenib [14]. 

The effect of celecoxib on phenotype switching has never been tested in melanoma before. The aim of this preliminary study was to evaluate if celecoxib could enhance the treatment response of melanoma cells to dabrafenib. In particular, we evaluated for the first time how celecoxib impacts on melanoma phenotype switching in vitro. The effect was tested on two human melanoma cell lines harboring the *BRAF^V600E^* mutation, A375 and SK-MEL-28, that are already in the MITF^low^/AXL^high^ end of the spectrum. 

## 2. Materials and Methods

### 2.1. Melanoma Cell Culture and Reagents

Two human melanoma cell lines were used: A375 (European Collection of Authenticated Cell Cultures, through Sigma Aldrich Company, St. Louis, MO, USA) and SK-MEL-28 (Cell line services, Eppelheim, Germany). Both melanoma cell lines were positive for *BRAF-V600E* mutation. Each cell line was seeded and maintained in Dulbecco’s modified Eagle Medium (DMEM), enriched with 10% fetal calf serum (FCS), 5 ng/mL amphotericin, and 50 μg/mL gentamicin, all reagents being from Biochrom AG, Berlin, Germany. The protocol followed standard culture conditions: cells were kept at constant temperature (37 °C) and humidity, under a 5% CO_2_ atmosphere, in an incubator, and were fed twice a week; all experiments were carried out in triplicate (*n* = 3); cells were used within a maximum of 4 passages. Dabrafenib and celecoxib were purchased from Cayman Chemical, MI, USA.

### 2.2. Cytotoxicity Assay

Viability tests were performed for each A375 (derived from a 54-year-old female) and SK-MEL-28 (derived from a 51-year-old male) cell line, as follows: melanoma cells were seeded in ELISA 96 wells micro titration flat bottom plates, at a density of 1 × 10^4^/well and incubated for 48 h, at 37 °C. Then, melanoma cells were exposed for 24–48–72–96 h either to dabrafenib, celecoxib, or the chosen combination. Viability tests were conducted separately with well-established concentrations of dabrafenib (ranging from 200, 300, 400, 500, 600, 700 nM) and celecoxib (ranging from 10, 20, 50, 100, 200, 250 nM). Stock solutions (1 mM) were prepared for each drug using dimethyl sulfoxide (DMSO) solution (Sigma-Aldrich, St. Louis, MO, USA). Serial dilutions were further obtained using cell culture medium. In addition, two concentrations were selected for each drug to be tested in combination for 72 h: celecoxib (C1 = 50 nM and C2 = 100 nM) and dabrafenib (D1 = 100 nM and D2 = 200 nM) dissolved in the medium. The final DMSO concentration used to prepare 50 nM celecoxib and 200 nM dabrafenib was 0.02%. Treated cells were compared to the control. Subsequent different exposure regimens, cytotoxicity assays, were performed by adding MTS and PMS solutions from CellTiter 96^®^ AQueous Non-Radioactive Cell Proliferation Assay (Promega Corporation, Madison, WI, USA) directly to wells. After 4 h of incubation, an ELISA plate reader (Tecan, Mannedorf, Switzerland) was used to measure the absorbance of formazan produced by living cells (540 nm). The results were presented as IC_50_.

### 2.3. Cell Death Mechanisms

Cell death mechanisms were analyzed at the end of the protocol (after 72 h). Treated melanoma cells were stained with Annexin V-fluorescein isothiocyanate (FITC)/vital dye propidium iodide (PI) (BD Pharmingen Biosciences, San Jose, CA, USA). Using a BD FACS Canto II flow cytometer (Becton Dickinson and Company, Franklin Lakes, NJ, USA) for flow cytometry, four groups were separated, as follows: viable cells (Annexin V (−)/PI (−)), early apoptotic cells (Annexin V-FITC (+)), late apoptotic cells (Annexin V (+)/PI (+)), and necrotic cells (PI (+)). The FACS analysis was completed using two lasers as excitation sources: blue (488 nm, air cooled, 20 mW solid state) and red (633 nm, 17 mWHeNe), as previously outlined. Forward scatter and side scatter were adjusted so that the majority of events appeared in the middle of the dot plot. A gate was drawn around those cells (P1) and only the events included within this gate were examined for FITC (annexin V) and PI signal. Debris was gated out by not including the events with a very low forward scatter (FSC below 5000) and that appeared at the bottom left corner of the dot plot. Moreover, discrimination of doublets was performed by plotting FSC-A (area) vs. FCS-H (height). Events that showed similar height as singlets but an increased area were gated out.

### 2.4. Structural Changes

Both melanoma cells were evaluated at the end of the protocol for structural changes using light microscopy, scanning electron microscopy (SEM), and transmission electron microscopy (TEM). 

#### 2.4.1. Light Microscopy

A375 and SK-MEL-28 melanoma cells’ response to treatment was evaluated during the protocol using an inverted microscope (Olympus CKX 41, Hamburg, Germany). Pictures were taken using a digital camera (Olympus, E 330, original magnification 10×).

#### 2.4.2. Scanning Electron Microscopy

The first set of cells from the two lines subjected to the various experimental treatments were grown on glass cover-slips placed at the bottom of the wells. At the end of the treatments, they were prefixed in ice-cold 2.7% glutaraldehyde (Agar Scientific, Stansted, UK) in 0.1 M phosphate buffer (at pH 7.4) for 1.5 h at 4 °C. The samples were washed with 0.1 M phosphate buffer (2 × 0.5 h at 4 °C), and postfixed with 1.5% OsO4 (Sigma-Aldrich, St. Louis, MO, USA) in 0.15 M phosphate buffer for 1.5 h at 4 °C. The cells were dehydrated with an ethanol (VWR International, Fontenay-sous-Bois, France) series of increasing concentrations (30% to absolute, 15 min each, at room temperature). After the cells complete drying, the cover-slips were placed on 10 mm/Ø9 mm aluminum stubs (Bio-Rad, Hercules, CA, USA), and the cells were sputter-coated with gold in a Polaron E-5100 sputter coater (Polaron Equipment Ltd., Watford, UK). Examination of samples was performed with a Jeol JSM-25S scanning electron microscope (Jeol Ltd., Tokyo, Japan), equipped with a Pixie 3000 system (Deben Ltd., Debenham, UK) for image acquisition.

#### 2.4.3. Transmission Electron Microscopy 

A second set of cells were detached from the culture plate by trypsinization for 4 min and centrifuged for 5 min at 500× *g*. The obtained pellets were redispersed in ice-cold 2.7% glutaraldehyde (Agar) in 0.1 M phosphate buffer (at pH 7.4), and the cells were prefixed for 1.5 h at 4 °C. The samples were washed with 0.1 M phosphate buffer (3 × 1 h, 1 × 24 h, all at 4 °C). Postfixation was performed next with 1.5% OsO4 (Sigma-Aldrich, St. Louis, MO, USA) in 0.15 M phosphate buffer for 1.5 h at 4 °C. The cells were further dehydrated with an acetone (International Laboratory, Cluj-Napoca, Romania) series of increasing concentrations (30% to absolute, 15 min each, at room temperature), and infiltrated with EMBED 812 (Electron Microscopy Sciences, Hatfield, PA, USA) (30% in acetone to pure resin 1 h each, at room temperature). After resin polymerization for 72 h at 60 °C, ultrathin sections of 60–80 nm were cut with a DiATOME diamond knife (DiATOME, Hatfield, PA, USA) on a LKB Ultrotome III Bromma 8800 ultramicrotome (LKB Produckter AB, Stockholm-Bromma, Sweden). The sections contrasted for 15 min with 13% uranyl acetate (Merck, Darmstadt, Germany) and for 5 min with 2.8% lead citrate (Fluka AG, Buchs, Switzerland) were examined with a JEOL JEM 1010 transmission electron microscope (Jeol Ltd., Tokyo, Japan), equipped with a Mega VIEW G2III camera and an iTEM software (Olympus, Soft Imaging System, Münster, Germany) for image acquisition.

### 2.5. Oxidative Stress Analysis

Malondialdehyde (MDA) levels, a marker for the peroxidation of membrane lipids, were measured by spectrophotometry using the fluorimetric method with 2-thiobarbituric acid (TBA), as described by Conti et al. All reagents were purchased from Sigma-Aldrich, St. Louis, MO, USA. Data were expressed as nmol/mg protein [15].

### 2.6. Cell Membrane Integrity Assay 

Lactate dehydrogenase (LDH), a cell membrane damage marker, was quantified from the melanoma culture medium using the spectrophotometric method. A mixture of 0.05 M sodium pyrophosphate buffer (pH 8.8), 5.25 × 10^−3^ M nicotineamide adenine dinucleotide (NAD), and 7.75 × 10^−2^ M lactate (final concentration) was added to each sample. All reagents were purchased from Abcam, Cambridge, UK. Absorbance was measured at 340 nm and data were expressed as U/mL [16].

### 2.7. Western Blot Analysis 

Key molecules were assessed via Western blot (WB), focusing on phenotype switching (transforming growth factor beta—TGF-β; microphthalmia transcription factor—MITF; the receptor tyrosine kinase AXL; Yes-associated protein 1—YAP1; transcriptional coactivator with PDZ-binding motif—TAZ), inflammation (cyclooxygenase-2—COX-2), and melanogenesis (tyrosinase). Following the 72-h protocol, cells were collected for each group and prepared for cell lysates according to the Bradford method (Bio-Rad, Hercules, CA, USA), considering bovine serum albumin as the standard. All lysates obtained were adjusted by total protein concentration. Cell lysates (20 μg protein/lane) were subsequently separated via electrophoresis on SDS PAGE gels and then transferred to polyvinylidene difluoride membranes using Bio-Rad Miniprotean system (Bio-Rad, Hercules, CA, USA). Protein migration was blocked using Blocking Buffer. The WB membranes were each incubated (16 h, at 4 °C) with primary antibodies against: MITF (C5) (1:500), AXL (H-3) (1:500), YAP1 (sc-101199) (1:500), TAZ (D-8) (1:500), COX-2 (H-3) (1:500), tyrosinase (H-109) (1:1000), GAPDH (FL-335) (1:1000) (Santa Cruz Biotechnology, Delaware Ave, Santa Cruz, CA, USA), and TGF-β (#3711) (1:1000) (Cell Signaling Technology, Cambridge, MA, USA). Membranes were washed and further incubated at room temperature, for 4 h, with corresponding secondary antibodies: anti-rabbit secondary antibody for MITF and tyrosinase; anti-mouse secondary antibody for TGF-β, AXL, YAP1, TAZ, COX-2, and GAPDH (Santa Cruz Biotechnology, Delaware Ave, Santa Cruz, CA, USA). Supersignal West Femto Chemiluminescent substrate (Thermo Fisher Scientific, Rockford, IL, USA), as well as Gel Doc Imaging system equipped with a XRS camera and Quantity One analysis software (Bio-Rad, Hercules, CA, USA) helped detect the proteins of interest. The protein-loading control was GAPDH from Trevigen Biotechnology Gaithersburg, MD, USA. The volume for each lane was quantified using Image Lab Software for PC Version 6.1, Bio-Rad, Hercules, CA, USA.

### 2.8. Statistical Method

All data were collected using Microsoft Excel for Windows 10 and then statistical analysis was performed using the one-way ANOVA, Tukey Post-tests and Nonlinear regression (curve fit) via GraphPad Prism version 8.00 for Windows, GraphPad Software, San Diego, CA, USA. Statistically significant results were considered when *p* value was less than 0.05 after comparing treated groups with the control. The suitable dose for the therapeutic combination was established after calculating the inhibitory concentration 50% (IC_50_) via Nonlinear regression (curve fit), dose–response inhibition. The panels illustrating the drug combination was generated using Excel for Microsoft 365 (Microsoft, Redmond, WA, USA).

## 3. Results

### 3.1. Cell Viability Analysis—IC_50_

The IC_50_ doses of celecoxib (C) and dabrafenib (D) drug combination were chosen after performing successive strategic viability tests on each melanoma cell line, A375 and SK-MEL-28. Viability quantified by colorimetry started using six different preset concentrations for each drug, evaluated for different exposure times (24 h, 48 h, 72 h, 96 h). Following our previous experience demonstrating that low doses of celecoxib can have an antineoplastic effect, we chose the same doses for celecoxib, 10, 20, 50, 100, 200, and 250 nM (Figure 1a). While treating our melanoma cells with low-doses of dabrafenib (1, 5, 10, 25, 50, 100 nM), we obtained a little reduction in cell viability, therefore, the final concentrations of dabrafenib were raised to 200, 300, 400, 500, 600, and 700 nM (Figure 1b). There was a direct proportionality between the drug dose and time exposure vs. cell viability. Two concentrations around IC_50_ were then chosen for both dabrafenib (D1 = 100 nM, D2 = 200 nM) and celecoxib (C1 = 50 nM, C2 = 100 nM) to be further tested as a combination (C + D) on each melanoma cell line, in quadruplicate (Figure 1c). As indicated in the drugs datasheet, the time exposure was set at 72 h for both celecoxib and dabrafenib. Subsequently, the dose of 50 nM celecoxib (A375: IC_50_ = 1.290 nM; SK-MEL-28: IC_50_ = 0.9957 nM) and 200 nM dabrafenib (A375: IC_50_ = 2.350 nM; SK-MEL-28: IC_50_ = 2.264 nM) (C1 + D2) was chosen for both melanoma cell lines for further assays.

### 3.2. Cell Death Mechanisms—FACS

Flow cytometry (fluorescence-activated cell sorting—FACS) was carried for both melanoma cell lines in order to assess the cell death mechanisms (Figure 2a–c. After the annexin/PI-staining of treated cells, the percentages of viable cells (Q3) compared to apoptotic cells (Q1 + Q2) from total cells were examined (b and c). 

The percentage of viable cells in the celecoxib group was not significantly different from the control, as celecoxib showed a reduced apoptotic effect on A375 cells compared to the control (Mean Diff. 0.5, 95% CI 0.2385 to 0.7615, *p* < 0.01). When tested in combination with dabrafenib, we observed a significant decrease in the percentage of viable cells in the celecoxib and dabrafenib group compared with the untreated cells (Mean Diff. 54.13, 95% CI 52.12 to 56.15, *p* < 0.0001) or the dabrafenib group (Mean Diff. 23.7, 95% CI 21.69 to 25.71, *p* < 0.0001). The main cell death mechanism was early apoptosis.

In SK-MEL-28 culture cells, celecoxib showed slight apoptosis induction compared to the control (Mean Diff. 0.5, 95% CI 0.2385 to 0.7615, *p* < 0.01), matching the viability tests and the microscopic evaluation. Regardless of these results, celecoxib effectively increased dabrafenib-induced cell death via apoptosis. SK-MEL-28 melanoma cells treated with the combined regimen underwent apoptosis in a significantly greater percentage compared to the control (Mean Diff.10.80, 95% CI 10.54 to 11.06, *p* < 0.0001) and dabrafenib alone (Mean Diff. 6.5, 95% CI 6.239 to 6.761, *p* < 0.0001), proving superior therapeutic efficacy. Moreover, different from A375 cell lines, there was a significant percent of necrotic cells observed in the dabrafenib and celecoxib + dabrafenib group. Additionally, judging by the total amount of viable cells left in the last group (77.2%), SK-MEL-28 seemed more resistant to treatment than A375.

### 3.3. Structural Changes—Light Microscopy, SEM, and TEM

#### 3.3.1. Light Microscopy

Both melanoma cell lines, A375 and SK-MEL-28, were exposed to a celecoxib and dabrafenib drug combination for 72 h. At the end of the protocol, the cells’ response to treatment was evaluated in each cell line with an inverted light microscope and pictures were taken for each group using a digital camera (Figure 3a,d). Even though there was a similar trend between the cell lines, A375 melanoma cells were more responsive to treatment than SK-MEL-28. In accordance to FACS results, the highest number of dead cells, small and rounded in shape, was observed in the last group (4. Celecoxib + Dabrafenib), followed by the dabrafenib group compared with the untreated cells (1. Control). Cell viability was less impaired in the celecoxib group compared to the control.

#### 3.3.2. Scanning Electron Microscopy (SEM)

A closer look was taken at the appearance of the cells using scanning electron microscopy—SEM (Figure 3b,e). Untreated A375 cells grew in enlarging, crowded groups of triangular, adherent cells, that showed a tendency to become more stretched, developing communication pseudopods with time. Untreated SK-MEL-28 melanoma cells were stellate, attached, with growing pseudopods and cell buds. In both melanoma cell lines, celecoxib did not much alter the cell appearance compared to the control group, except the presence of rare, rounded, dead cells. The dabrafenib treatment induced significant melanoma cell death. Most of the cells were smaller, detached, and rounded, in general found in groups. However, a few elongated, viable cells remained, with lost pseudopods. The celecoxib and dabrafenib treatment combination even further reduced the number of viable cells in the A375 line, visible as isolated or, more often, detached groups of globular cells. Viable cells left in the last group in the SK-MEL-28 line appeared faded.

#### 3.3.3. Transmission Electron Microscopy (TEM)

Intracellular changes were also tracked in treated A375 and SK-MEL-28 melanoma cells using transmission electron microscopy—TEM (Figure 3c,f). A normal-appearing nucleus with prominent nucleoli, surrounded by intracytoplasmic organelles could be observed in untreated melanoma cells. In addition to this general aspect of A375 melanoma cells, SK-MEL-28 cells presented little branched extensions at the cell surfaces. Melanoma cells treated with dabrafenib underwent vacuolic degeneration inside the cytoplasm, as well as sparse chromatin condensation inside the nucleus. SK-MEL-28 cells lost the pericellular buds seen initially in the control group. In the celecoxib group, melanoma cells developed a few vacuoles inside the cytoplasm. Taking a closer look, we observed normal-appearing mitochondria, as well as mitochondria in different stages of ballonization. This observation raised the suspicion that vacuoles seen in the celecoxib + dabrafenib group might represent destroyed mitochondria. In the group treated with both celecoxib and dabrafenib, we observed severely destroyed cells, filled with numerous vacuoles, with no visible organelles and severe nuclear alterations.

### 3.4. Oxidative Stress Analysis—MDA

Lipid oxidation gives rise to malondialdehyde production (MDA). It is well known that both dabrafenib and celecoxib increase oxidative stress in targeted cells after upregulating reactive oxygen species (ROS) and MDA levels in a dose-dependent manner. Melanoma cells were subjected to a celecoxib and dabrafenib therapeutic combination for 72 h and MDA levels were evaluated as a biological marker of oxidative stress, as shown in Figure 4a,b. In both melanoma cell lines, we observed the same pattern. When compared to the control, the highest MDA levels were observed after the celecoxib and dabrafenib combination (Mean Diff. 0.9114, 95% CI 0.8609 to 0.9620, *p* < 0.0001 for A375; Mean Diff. 0.5730, 95% CI 0.5381 to 0.6079, *p* < 0.0001 for SK-MEL-28). Results suggest that both celecoxib and dabrafenib increase oxidative stress, resulting in significantly higher MDA levels in the last group compared to the dabrafenib group (Mean Diff. 0.4527, 95% CI 0.4021 to 0.5032, *p* < 0.0001 for A375; Mean Diff. 0.09487, 95% CI 0.0599 to 0.1298, *p* < 0.0001 for SK-MEL-28). 

### 3.5. Cell Membrane Integrity Assay—LDH

LDH is a cytosolic enzyme released into the cell culture medium after damage of the plasma membrane. The LDH level in the spent culture medium is inversely proportional to the viable cell population with intact membrane integrity in the culture. Following the different exposure regimens, the medium was collected and analyzed for LDH levels for both A375 and SK-MEL-28 melanoma cell lines (Figure 5a,b). The highest LDH levels were obtained in the celecoxib + dabrafenib group compared to the control group (Mean Diff. 3.767, 95% CI 3.336 to 4.197, *p* < 0.0001 for A375; Mean Diff. 4.54, 95% CI 4.063 to 5.016, *p* < 0.0001 for SK-MEL-28), as well as the dabrafenib group (Mean Diff. 1.975, 95% CI 1.545 to 2.406, *p* < 0.0001 for A375; Mean Diff. 1.472, 95% CI 0.9958 to 1.949, *p* < 0.0001 for SK-MEL-28). Results correlate with the cell viability and FACS results.

### 3.6. Western Blot Analysis

Following different treatment regimens, the expression of key proteins involved in the Hippo pathway were further analyzed via Western Blot tests in A375 (Figure 6) and SK-MEL-28 (Figure 7) melanoma cells, respectively. Specifically, we focused on evaluating how TGF-β [17], MITF [13], AXL [18], YAP1 [19], TAZ [20], COX-2 [21], and tyrosinase [13] expression is influenced by celecoxib and dabrafenib treatment.

**TGF-β**. TGF-β release controls extracellular matrix stiffness (ECM), and promotes EDNRA receptor activation and nuclear YAP1 expression with further intracellular AXL production. This cascade is used by melanoma cells when switching to a de-differentiated/invasive state, resistant to RAF/MEK inhibitors. We observed the same pattern in both the *BRAF* mutant melanoma cells. Surprisingly, the highest levels of TGF-β were obtained in the celecoxib group. Low doses of celecoxib increased the TGF-β release compared to the untreated cells in both A375 (Mean Diff.1.314, 95% CI 1.252 to 1.376, *p* < 0.0001) and SK-MEL-28 (Mean Diff. 0.1932, 95% CI 0.1281 to 0.2583, *p* < 0.0001) melanoma cells. The celecoxib and dabrafenib treatment combination increased TGF-β expression compared to untreated cells in the A375 cell line (Mean Diff. 0.6832, 95% CI 0.6208 to 0.7455, *p* < 0.0001), while in SK-MEL-28, we obtained lower levels of TGF-β in the last group compared to the control (Mean Diff. 0.08453, 95% CI 0.01941 to 0.1496, *p* < 0.05). Dabrafenib significantly decreased TGF-β expression compared to the control group (Mean Diff. 0.3095, 95% CI 0.2471 to 0.3718, *p* < 0.0001 in A375; Mean Diff. 0.4277, 95% CI 0.3626 to 0.4928, *p* < 0.0001 in SK-MEL-28) and the celecoxib + dabrafenib group (Mean Diff. 0.9926, 95% CI 0.9303 to 1.055, *p* < 0.0001 in A375; Mean Diff. 0.3432, 95% CI 0.2781 to 0.4083, *p* < 0.0001 in SK-MEL-28) in both melanoma cells. Results suggest that adding celecoxib to dabrafenib reduces the therapeutic response of melanoma cells.

**MITF**. Phenotype switching in melanoma is of crucial importance when talking about resistance to BRAF inhibitors. The most important phenomenon is represented by the MITF expression that correlates inversely with AXL levels. The highest MITF expression was obtained in the dabrafenib group in both A375 (Mean Diff. 1.396, 95% CI 1.135 to 1.658, *p* < 0.0001 compared to the control; Mean Diff. 0.5462, 95% CI 0.2847 to 8078, *p* < 0.001 compared to the fourth group) and SK-MEL-28 (Mean Diff. 1.376, 95% CI 1.187 to 1.565, *p* < 0.0001 compared to the control; Mean Diff. 0.9746, 95% CI 0.7858 to 1.163, *p* < 0.0001 compared to the fourth group) melanoma cells. There was no difference in MITF expression between the celecoxib group and untreated cells (*p* > 0.05). As a result, the last group expressed increased levels of MITF compared to the control (Mean Diff. 0.8502, 95% CI 0.5887 to 1.112, *p* < 0.0001 in A375 group; Mean Diff. 0.4015, 95% CI 0.2127 to 0.5903, *p* < 0.001 in SK-MEL-28), but lower levels compared to the dabrafenib group (Mean Diff. 0.5462, 95% CI 0.2847 to 0.8078, *p* < 0.001 in A375 group; Mean Diff. 0.9746, 95% CI 0.7858 to 1.163, *p* < 0.0001 in SK-MEL-28).

**AXL**. AXL expression was recently associated with motility and invasion. Melanoma tumors expressing MITF^low^/AXL^high^ levels adopt a de-differentiated/invasive state and become resistant to targeted therapies. In A375 melanoma cells treated with dabrafenib, we obtained the lowest AXL levels, lower than the control group (Mean Diff. 0.2960, 95% CI 0.2248 to 0.3671, *p* < 0.0001) and lower than the last group of interest (Mean Diff. 1.292, 95% CI 1.220 to 1.363, *p* < 0.0001). The highest AXL level was obtained in the celecoxib group compared with the control (Mean Diff. 1.177, 95% CI 1.106 to 1.248, *p* < 0.0001). As a result, AXL levels were significantly increased in the last group compared to the control (Mean Diff. 0.9956, 95% CI 0.9244 to 1.067, *p* < 0.0001). In SK-MEL-28, there was no difference in AXL expression between the first two groups. In this melanoma cell line, celecoxib raised AXL expression in the last group as well compared to the control (Mean Diff. 0.4910, 95% CI 0.4677 to 0.5143, *p* < 0.0001) or dabrafenib group (Mean Diff. 0.4735, 95% CI 0.4501 to 0.4968, *p* < 0.0001). Overall, celecoxib decreased therapeutic efficacy, contributing to a tendency of MITF^low^/AXL^high^ expression in both melanoma cell lines compared to the dabrafenib group.

**YAP1**. When Hippo signaling is reduced in cancer this leads to the accumulation of YAP1/TAZ complexes in the nucleus in order to promote cell proliferation and survival. Increased oncoprotein YAP1 levels facilitate increased AXL expression. In the A375 melanoma cell line, dabrafenib strongly inhibited YAP1 expression (Mean Diff. 0.6259, 95% CI 0.5941 to 0.6578, *p* < 0.0001), while the celecoxib group registered the highest levels of YAP1 (Mean Diff. 0.05181, 95% CI 0.01994 to 0.8367, *p* < 0.01) compared with the control group. The therapeutic combination had a higher YAP1 expression than the dabrafenib group (Mean Diff. 0.4642, 95% CI 0.4323 to 0.4961, *p* < 0.0001), but lower than the control (Mean Diff. 0.1617, 95% CI 0.1298 to 0.1936, *p* < 0.0001). YAP1 results explain AXL expression. In the SK-MEL-28 melanoma cell line, dabrafenib also showed the highest inhibitory effect on YAP1 compared to the control (Mean Diff. 0.1317, 95% CI 0.1149 to 0.1485, *p* < 0.0001). The therapeutic combination had the highest YAP1 expression compared to the untreated cells (Mean Diff. 0.1276, 95% CI 0.1108 to 0.1444, *p* < 0.0001) or the dabrafenib group (Mean Diff. 0.2593, 95% CI 0.2425 to 0.2761, *p* < 0.0001), correlating with AXL levels as well.

**TAZ**. YAP1/TAZ are not only essential for cancer initiation, but they also function as sensors of the mechanical or structural features of the cell microenvironment. In both cell lines, the highest levels of TAZ were observed in the group of the celecoxib and dabrafenib combination compared to the control (Mean Diff. 1.397, 95% CI 1.316 to 1.479, *p* < 0.0001 for A375; Mean Diff. 0.398, 95% CI 0.3738 to 0.4221, *p* < 0.0001 for SK-MEL-28). Furthermore, in each case, celecoxib raised TAZ expression compared to the control (Mean Diff. 1.295, 95% CI 1.214 to 1.376, *p* < 0.0001 for A375; Mean Diff. 0.2621, 95% CI 0.2379 to 0.2863, *p* < 0.0001 for SK-MEL-28) or dabrafenib group, suggesting once again that celecoxib strongly promotes YAP1/TAZ activation with AXL production. 

**COX-2.** YAP1 transcriptionally regulates COX-2 expression, an important enzyme for the chronic inflammatory state in melanoma. In both melanoma cell lines, we observed the same pattern. The lowest COX-2 expression was obtained in the celecoxib group, followed by the control and the combined regimen. Dabrafenib registered the highest levels of COX-2 expression compared to the group of interest in A375 (Mean Diff. 11.96, 95% CI 11.80 to 12.11, *p* < 0.0001) and SK-MEL-28 (Mean Diff. 0.3227, 95% CI 0.2985 to 0.3468, *p* < 0.0001). COX-2 expression in the last group was significantly higher than the control in each cell line (Mean Diff. 4.327, 95% CI 4.172 to 4.481, *p* < 0.0001 for A375; Mean Diff. 0.7662, 95% CI 0.7135 to 0.8188, *p* < 0.0001 for SK-MEL-28).

**TYROSINASE**. The overexpression of the tyrosinase enzyme as a marker of melanin production is frequently observed in highly aggressive melanocytic tumors. Tyrosinase activity increased under dabrafenib treatment compared to untreated cells in both A375 (Mean Diff. 2.538, 95% CI 2.439 to 2.637, *p* < 0.0001) and SK-MEL-28 (Mean Diff. 7.275, 95% CI 6.833 to 7.717, *p* < 0.0001) cell lines. Even though the therapeutic combination reduced tyrosinase expression compared to the dabrafenib group, its expression was still higher than the control in both cell lines (Mean Diff. 0.2651, 95% CI 0.166 to 0.3641, *p* < 0.001 for A375; Mean Diff. 5.010, 95% CI 4.568 to 5.452, *p* < 0.0001). These results suggest that celecoxib and dabrafenib therapeutic combination might rather fail to inhibit melanogenesis in these melanoma cell lines.

## 4. Discussion

The presence of the activating *BRAF^V600E^* mutation in more than 50% of cutaneous melanomas has forced the discovery of BRAF and MEK inhibitors, called targeted therapies. Although at first, the combination of targeted inhibitors reduces the tumor burden and prolongs the patients’ overall survival by more than 2 years, eventually, melanomas develop resistance and restart progressing while on treatment [22,23]. Many resistance mechanisms have been depicted, highlighting how complex melanoma biology actually is [7]. Among them, phenotype switching, represented by a low MITF/AXL ratio, has gained a lot of attention [24]. In a previous work, we showed that low-dose celecoxib exerts antineoplastic effects in a melanoma co-culture treated with trametinib, by decreasing phosphatidylinositol 3-kinase/protein kinase B (PI3K/AKT) resistance pathway activation [14]. Moreover, what is already known about the antineoplastic capabilities of celecoxib in melanoma, independent from COX-2/prostaglandin E2 (PgE_2_) inhibition, is that it suppresses cancer cell growth and promotes apoptosis via Wnt/β-catenin and m-TOR (mechanistic target of rapamycin) substitutive pathway inhibition, and reduces the epidermal growth factor receptor (EGFR), vascular endothelial growth factor (VEGF), and the signal transducer and activator of transcription 3 (STAT3) expression [25,26,27,28]. 

This study evaluated, for the first time, the chemopreventive effect of celecoxib in overcoming the phenotype switching phenomenon in two metastatic melanoma cell lines encoding the *BRAF*^V600E^ mutation. The therapeutic association of celecoxib (50 nM) and dabrafenib (200 nM) was tested in vitro for 72 h on SK-MEL-28 and A375 melanoma cell lines. According to the literature and suggested by our own WB results, SK-MEL-28 and A375 melanoma cell lines are already in the MITF^low^/AXL^high^ end of the spectrum [29]. We were particularly interested in the evaluation of the celecoxib and dabrafenib synergism, how they impact on cell viability and oxidative stress-related cell death, phenotype switching, inflammation, melanogenesis, and correlated with structural changes. 

Starting from viability tests we observed that melanoma cells manifested sustained resistance to usual doses of dabrafenib (1–100 nM—data not shown), therefore, we had to increase dabrafenib doses in order to achieve IC_50_ (200–700 nM) [30]. Similar to our previous expertise, celecoxib did not alter cell viability as a single compound [14]. However, the celecoxib and dabrafenib combination induced a time and dose-dependent cytotoxic effect in both A375 and SK-MEL-28 cell lines. Celecoxib potentiated dabrafenib’s cytotoxic capabilities after 72 h. There was a clear cell response to the combined treatment in A375, apoptosis being the main cell death mechanism observed after FACS analysis. SK-MEL-28 melanoma cells proved to be more resistant to the therapeutic combination, with cells undergoing both apoptosis and necrosis. LDH levels were measured, indicating cell membrane damage and induced cancer cell death as a cumulative inhibitory effect [31]. 

Oxidative stress was related to melanoma cells’ resistance to BRAFi. On one hand, BRAFi treatment represented by dabrafenib induced ROS production, allowing metabolic processes to rewire and influence melanoma cells’ growth [32]. On the other hand, celecoxib exerts its antitumoral capabilities by promoting high reactive oxygen species (ROS) and MDA production in a dose-dependent manner, while inhibiting mitochondrial O_2_ consumption [33,34]. Overall, the highest MDA levels obtained in the celecoxib + dabrafenib group as a sign of cumulative ROS production in A375 and SK-MEL-28 melanoma cells might rather represent a sign of treatment resistance [35]. FACS analysis following celecoxib and dabrafenib treatment was unanimous with the LDH and MDA results. All these changes were objectified using light microscopy, SEM, and TEM in order to track extracellular and intracellular changes. The cytotoxic effect of the combined regimen marked cell appearance in the last group where the highest number of dead cells was observed: smaller and rounded in shape, detached from the well, suffering mitochondria ballonization and vacuolic degeneration inside the cytoplasm, with sparse chromatin condensation inside the nucleus, until the complete damage of the melanoma cells. These changes were highlighted for the first time as an effect of celecoxib associated with dabrafenib. 

The Hippo signaling pathway tactfully controls the proliferation vs. apoptosis balance. The pathway is extracellularly regulated by diffusible chemicals, mechanical stimuli, and signals sensed by G-protein coupled receptors (for instance endothelin receptor type A—EDNRA, endothelin receptor type B—EDNRB). The Hippo signaling pathway also supervises organ size in animals after negatively regulating the expression of YAP/TAZ downstream oncoproteins [7]. The active Hippo pathway prevents cell proliferation by keeping YAP/TAZ phosphorylated into the cytoplasm. The Hippo pathway is often inactivated in cancers and leads to cell division and the inhibition of apoptosis after dephosphorylated YAP/TAZ translocation to the nucleus [36]. Matrix rigidity also promotes nucleus localization and the activity of YAP. In cancer-associated fibroblasts, YAP also acts as a critical factor in determining ECM remodeling via TGF-β secretion, towards increased stiffening [37]. The biology of YAP/TAZ is very impressive, as they have the capability to reprogram cancer cells into cancer stem cells, standing at the roots of cancer [38]. Elevated YAP expression was associated with resistance to BRAF/MEK inhibitors and a poor prognosis [39]. YAP/TAZ activation may also induce resistance to radiotherapy, chemotherapy, or immunotherapy [40]. Recent in vitro studies showed that YAP was sufficient for melanoma cell invasion. Moreover, YAP overexpression switched the melanoma cells’ phenotype from proliferative to invasive by driving the expression of AXL, thrombospondin1 (THBS1), and cysteine-rich angiogenic inducer 61 (CYR61) [41,42]. Interestingly, melanoma cells can switch between invasive and proliferative states because they suffer cellular transcriptome changes, downstream of signaling events. For instance, BRAFi drive melanoma cells to adopt a resistant state after MITF^low^ and AXL^high^ expression, the prototype for a de-differentiated/mesenchymal/invasive phenotype. Thus, in aggressive melanoma tumors with acquired resistance, MITF and AXL levels correlate inversely [24]. Melanoma tumors have such a complex microenvironment characterized by many reversible phenotypic switches, driving in the end MITF^low^ (de-differentiated/invasive) and MITF^high^ (differentiated/proliferative) cells to coexist. It seems that the MITF modulates the production of an inflammatory secretome and the expression of programmed cell death—ligand 1 (PD-L1). In this manner, the MITF directly controls the infiltration and activation of the immune cells [43]. Müller et al. have emphasized that the lack of a MITF is linked with a more severe resistance to targeted drugs, while its presence is needed for a robust drug response. Furthermore, the MITF and tyrosinase enzyme are responsible for pigmentation in both nevi and melanoma cells with a proliferative phenotype [24]. These data indicated a therapeutically exploitable perspective for the Hippo pathway and phenotype switching in melanoma.

The celecoxib and dabrafenib drug combination tested on two aggressive melanoma cell lines, A375 and SK-MEL-28, followed almost the same pattern. The highest levels of TGF-β, suggestive for ECM stiffness, were obtained in the celecoxib group compared to the control, while TGF-β was unanimously inhibited by dabrafenib in both cell lines. Even though A375 showed a better therapeutic response with significantly increased cell death after the combined regimen (44% viable cells), this activated a TGF-β release in the last group greater than the control and dabrafenib group. The percent of viable SK-MEL-28 cells was not that impaired in the last group (77% viable cells) and TGF-β levels were significantly lower in the fourth group compared to the control. Increased TGF-β makes the switch from the EDNRB G-protein-coupled receptor-mediated MITF and tyrosinase expression to YAP/TAZ activation via EDNRA G-protein-coupled receptors, promoting AXL production [7]. In both A375 and SK-MEL-28, there was no difference in the MITF expression between the control and celecoxib group, while dabrafenib significantly increased MITF expression alone and in the last group. The MITF results inversely correlated with AXL levels. Celecoxib significantly increased AXL levels in the last group compared to the control and dabrafenib used alone. These results were consistent with high YAP1/TAZ levels in the last group compared to the control in both melanoma cell lines. 

The correlation between COX-2 expression and the tendency to growth and invasion is still partially understood in metastatic melanoma [11]. In that respect, COX-2 expression was strongly inhibited by the combined regimen compared to dabrafenib alone, but the levels were still high compared to untreated cells. An explanation could be that BRAF-mutated melanoma cells express high levels of COX-2 [44]. Moreover, increased doses of dabrafenib might paradoxically increase COX-2 expression, while low doses of dabrafenib (25 nM) do not impact on COX-2 expression [45]. Celecoxib inhibited COX-2 expression in both cell lines. However, COX-2 levels did not show the same trend as YAP1 in our study [42]. Maybe this is because subpopulations of melanoma cells with a different phenotypic status coexist in the same microenvironment. Since celecoxib’s antineoplastic effects are also COX-2 independent, we considered this cannot exclude its influence on phenotype switching [11]. These observations reminded us how intricate melanoma biology actually is. Inflammatory markers, such as COX-2, deserve further investigation in order to understand their position in phenotype switching phenomenon in advanced melanoma biology.

Melanogenesis, measured by tyrosinase levels, was significantly reduced in the last group compared to dabrafenib used alone, but remained higher than the levels expressed in untreated cells. These results confirm the rule that low pigmentation and reduced proliferation corresponds to invasion and metastasis formation [46].

From the beginning, we preferred a simple experimental design to evaluate the impact of celecoxib on phenotype switching ex vivo, in melanoma. Our results might rather indicate a lack of synergism between celecoxib and dabrafenib regarding phenotype switching in vitro. Even though celecoxib association to dabrafenib inhibited cell viability and promoted oxidative stress-mediated cell death, surprisingly, celecoxib reduced the antineoplastic effect of dabrafenib, promoted ECM stiffness, induced YAP1/TAZ expression, and reduced the MITF/AXL ratio. These findings remind us that melanoma is a versatile tumor and its resistance mechanisms need to be evaluated as a whole. Overall, celecoxib seemed to inhibit the Hippo pathway and promote the switch to a mesenchymal/de-differentiated/invasive phenotype in A375 and SK-MEL-28 melanoma cells.

Breast cancer cells are also characterized by these dynamic morphological phenotypes in response to the local social environment [47]. An interesting trial (REMAGUS02 Trial) conducted on breast cancer patients warned that celecoxib associated with neoadjuvant chemotherapy might worsen outcomes differentially by COX-2 expression [48]. This idea might be relevant for melanoma patients as well.

One also has to be cautious that these are only preliminary results. Our findings require further validation in other melanoma cell lines, in co-cultures and on animal models. Another limitation would be that low doses of celecoxib were used for the protocol. It would be interesting to evaluate if celecoxib is still able to promote the invasive phenotype in higher doses (μM range).

## 5. Conclusions

There is reticence about the use of celecoxib as a chemopreventive agent in melanoma clinical studies before depicting melanoma biology and its resistance mechanisms. Our preliminary study emphasized there is still much to learn about melanoma microenvironment interactions and how celecoxib exerts its antineoplastic effects in melanoma. Low doses of celecoxib might inhibit the Hippo pathway and promote an invasive phenotype via AXL^high^/MITF^low^ expression in A375 and SK-MEL-28 melanoma cells. These findings clearly need further in vitro and in vivo confirmation and could represent a hypothesis that partially explains the lack of clinical validation of celecoxib in melanoma.

## Figures and Tables

**Figure 1 jcm-11-04560-f001:**
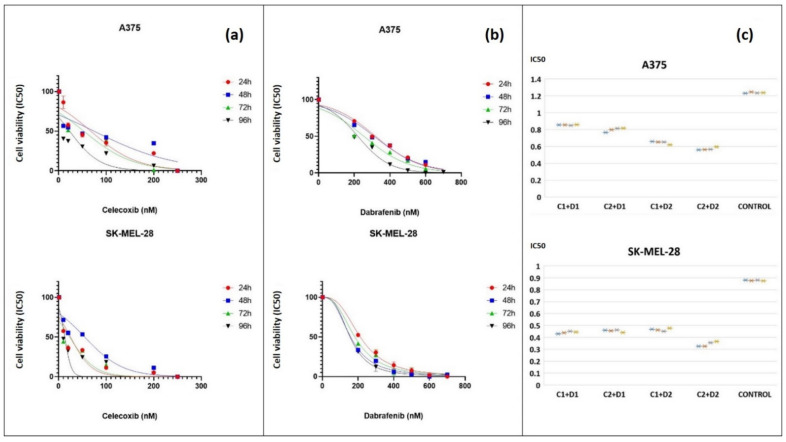
Cell viability analysis after exposure to celecoxib and dabrafenib. Experiments started with A375 and SK-MEL-28 melanoma cell lines exposed for 24, 48, 72, and 96 h to different concentrations of celecoxib (**a**) and dabrafenib (**b**). Following these viability tests two concentrations were selected for celecoxib (C1 = 50 nM, C2 = 100 nM) and dabrafenib (D1 = 100 nM, D2 = 200 nM) to be evaluated as a novel drug combination for 72 h (**c**). The first two IC_50_ graphs (**a**,**b**) were obtained using GraphPad Software, Nonlinear regression (curve fit) and show mean and error values ± standard deviation (SD), *n* = 3 for each sample. The third panel (**c**) illustrates cell viability analysis after 72 h exposure to four drug combinations using Excel-Box and Whisker. Each drug combination was tested in quadruplicate (*n* = 4) and was marked as asterisk (*) on the graph. Cell viability diminished with increasing concentrations of celecoxib and dabrafenib in a dose- and time-dependent manner. The suitable IC_50_ doses were chosen for celecoxib (50 nM) (A375: IC_50_ = 1.290 nM, R^2^ = 0.8549; SK-MEL-28: IC_50_ = 0.9957 nM, R^2^ = 0.9530) and dabrafenib (200 nM) (A375: IC_50_ = 2.350 nM, R^2^ = 0.9690; SK-MEL-28: IC_50_ = 2.264 nM, R^2^ = 0.9777) to be further used as a combination in melanoma.

**Figure 2 jcm-11-04560-f002:**
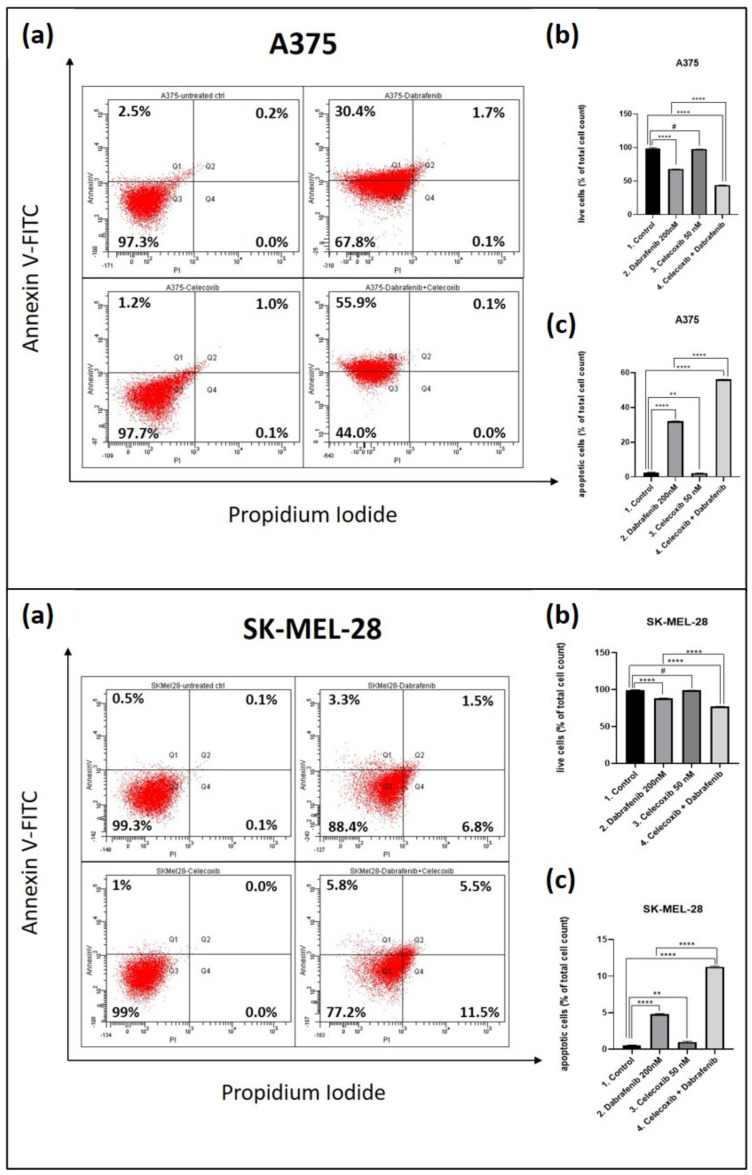
FACS cell death assessment. Comparative FACS analysis following celecoxib and dabrafenib combination was carried in A375 (**upper panel**) and SK-MEL-28 (**lower panel**) human melanoma cell lines. Quantitative FACS results in each case are illustrated as % of total cell count of annexin V- and PI-positive cells, compared to total cell count (**a**). Viable cells (**b**) and Q1 + Q2 apoptotic cells (**c**) were statistically analyzed using GraphPad Software, Ordinary One-way ANOVA test, Multiple comparisons. There was no statistical difference between the viable untreated cells and those treated with celecoxib, although celecoxib induced a slight apoptotic effect. However, celecoxib added to dabrafenib enhanced apoptosis-mediated cell death in the last group compared to control and dabrafenib group. # = *p* > 0.05 (not significant), ** *p* < 0.01, **** *p* < 0.0001 vs. control, untreated cells and **** *p* < 0.0001 vs. dabrafenib group. Each bar represents mean values ± standard deviation (SD), *n* = 3.

**Figure 3 jcm-11-04560-f003:**
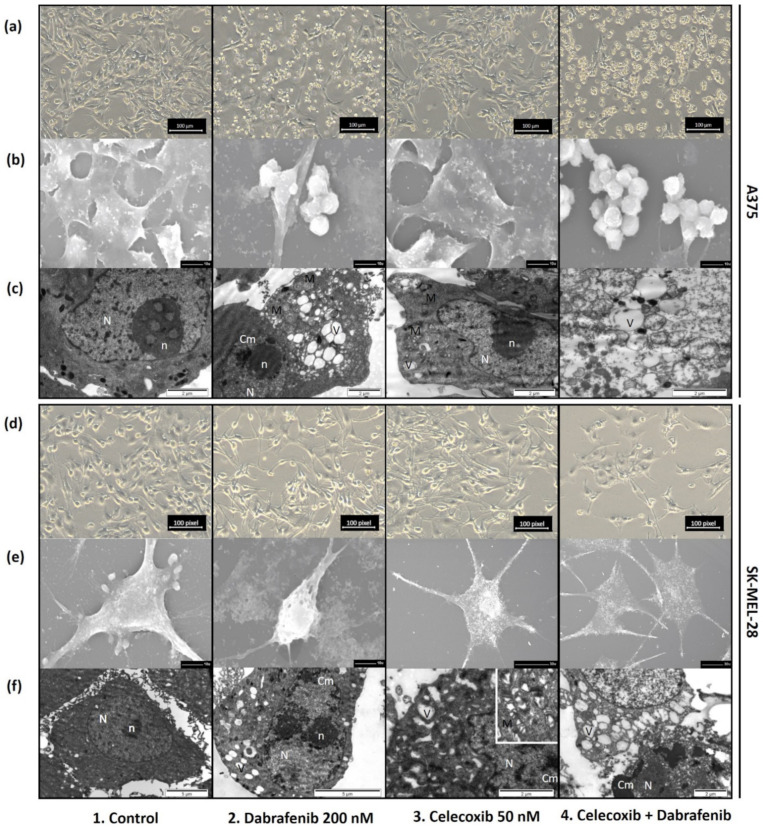
External and internal melanoma cells change after 72 h of celecoxib and dabrafenib treatment. During treatment, cells were observed using an inverted light microscope (Olympus CKX41, Hamburg, Germany) and pictures were taken at the end of each protocol using a digital camera (Olympus E330) (**a**,**d**). Moreover, some cells were seeded on glass coverslips in order to be further evaluated using SEM (**b**,**e**) and cells from the supernatant were collected and used for evaluation of internal changes using TEM (**c**,**f**). Scale bars = 100 µm (**a**,**d**), 10 µm (**b**,**e1**,**e2**), 2 µm (**c**,**f3**,**f4**), 30 µm (**e3**,**e4**), 5 µm (**f1**,**f2**). N, nucleus; *n*, nucleolus; V, vacuoles; Cm, chromatin; M, mitochondria.

**Figure 4 jcm-11-04560-f004:**
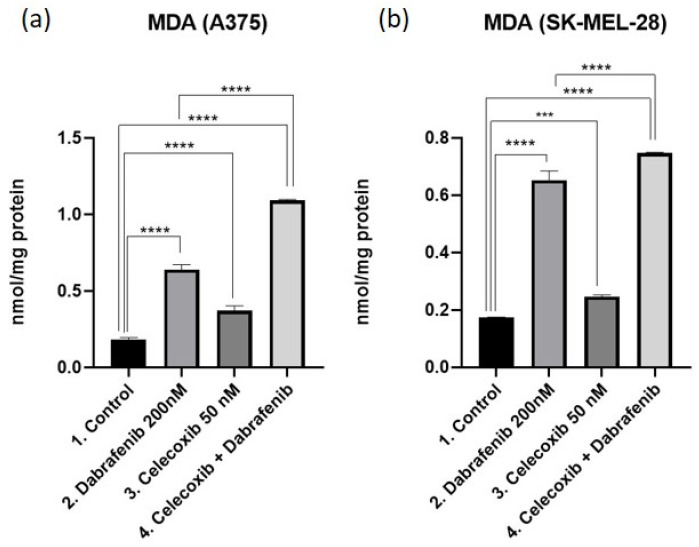
MDA as a marker for oxidative stress. MDA measurements in A375 (**a**) and SK-MEL-28 (**b**) were done by spectrophotometry. Results were statistically analyzed using GraphPad Software, Ordinary One-way ANOVA test, Multiple comparisons. Each bar represents mean ± standard deviation, *n* = 3. *** = *p* < 0.001, **** = *p* < 0.0001.

**Figure 5 jcm-11-04560-f005:**
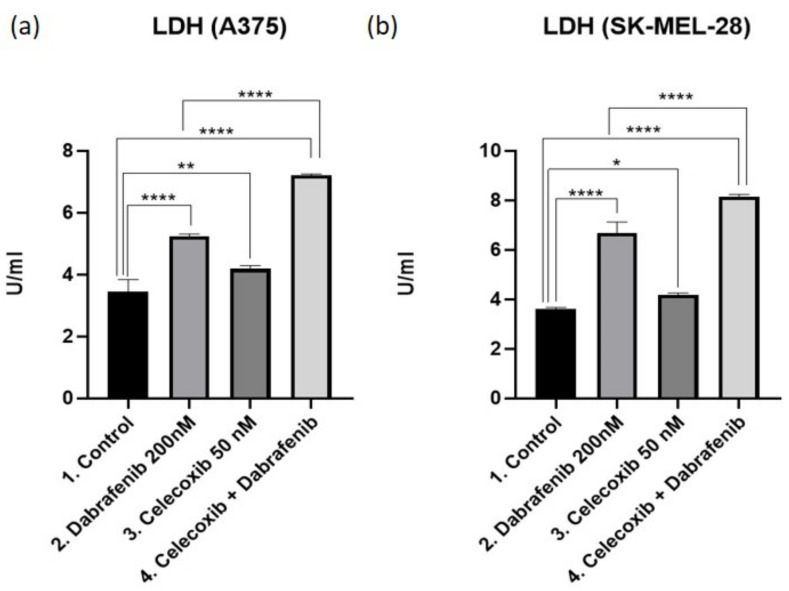
LDH as a marker for cell membrane integrity. LDH levels were measured in both A375 (**a**) and SK-MEL-28 (**b**) cell culture medium as a result of the cytotoxic effect of the drugs used. Results were statistically analyzed using GraphPad Software, Ordinary One-way ANOVA test, Multiple comparisons. Each bar represents mean ± standard deviation, *n* = 3. * = *p* < 0.05, ** = *p* < 0.01, **** = *p* < 0.0001.

**Figure 6 jcm-11-04560-f006:**
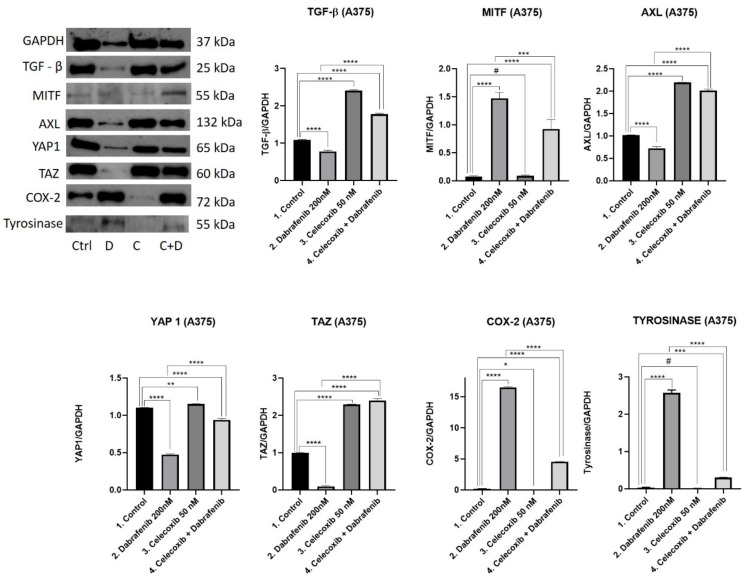
Western blot analysis for A375. Results were analyzed using GraphPad Software, Ordinary One-way ANOVA test, Multiple comparisons. Each bar represents mean ± standard deviation, *n* = 3. # = *p* > 0.05 (not significant), * = *p* < 0.05, ** = *p* < 0.01, *** = *p* < 0.001, **** = *p* < 0.0001.

**Figure 7 jcm-11-04560-f007:**
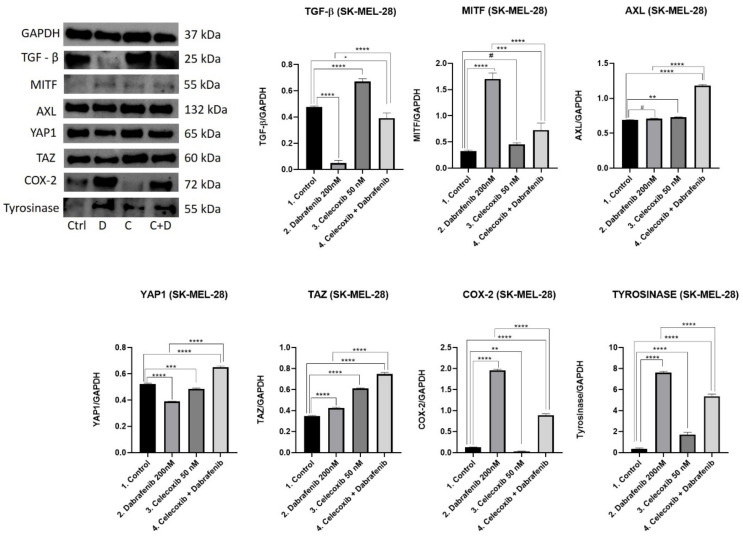
Western blot analysis for SK-MEL-28. Results were statistically analyzed using GraphPad Software, Ordinary One-way ANOVA test, Multiple comparisons. Each bar represents mean ± standard deviation, *n* = 3. # = *p* > 0.05 (not significant), * = *p* < 0.05, ** = *p* < 0.01, *** = *p* < 0.001, **** = *p* < 0.0001.

## Data Availability

The data presented in this study are openly available at doi. The data that support the findings of this study are available from the corresponding author, D.V.T., upon reasonable request.

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
