# Peer review of "Low Doses of Celecoxib Might Promote Phenotype Switching in Cutaneous Melanoma Treated with Dabrafenib—Preliminary Study"

_jcm, 2022, doi:10.3390/jcm11154560_

Round 1
Reviewer 1 Report
Dear Authors:
I reviewed with great interest the manuscript entitled “Low doses of Celecoxib might promote phenotype switching in cutaneous melanoma treated with Dabrafenib – preliminary study”, jcm-1814168 by Tudor et al. The investigators evaluate the effects of celecoxib alone and in combination with dabrafenib in two resistant melanoma cell lines. Of note, the authors assessed for the first-time celecoxib efficacy with regard to phenotypic switching in melanoma. Interestingly, when the expression of important proteins involved in phenotypic switching are detected following drug treatment, a lack of synergy was detected between celecoxib and dabrafenib. This result is seemingly paradoxical to the detected increased cell death when the drugs were used in combination. Despite initial cell death, celecoxib may inhibit the Hippo pathway, and the authors hypothesize that this could lead to a more invasive phenotype via increase MITFLow/AXLHigh expression. This preliminary data has important implications for the enhancement of our understanding of the complexity of melanoma drug resistance as well as to provide support for the lack of clinical validation of celecoxib in melanoma. However, there are several considerations that should be addressed. Below are comments directed toward minor suggested grammatical changes and well as some major suggested changes for figures and data interpretation.
Introduction:
The authors provide sufficient background on the importance of understanding mechanisms of resistance in melanoma and in particular the role phenotypic switching in melanogenesis. However, the importance of celecoxib as a melanoma therapeutic is lacking. Additional references should be added to this section starting at line 64. This will strengthen the importance of this study to their audience up front. It would also be beneficial to the reader if there was a statement or two about why the authors chose to evaluate celecoxib and dabrafenib in combination and not another well-known targeted therapeutic for melanoma such as vemurafenib or trametinib.
Line 37: Change “having” to “have”
Line 41: Add “for” to the sentence; “advanced stages for melanoma patients.”
Line 42: Change “were” to “are” in the sentence; “There are described several ways…”
Line 53: Add “and”; “matrix, and regulates starvation-induced autophagy.”
Line 64: Add “is”; “Cycoloxygenase-2 (COX-2) overexpression is correlated…”
Line 66: Add “at”; “Even at low doses…”
Line 66 and throughout the manuscript: be consistent with not capitalizing celecoxib when not at the start of a sentence.
Line 67: It would be beneficial for the reader to state what the melanoma cells were co-cultured with in reference XII.
Methods:
Overall, the methods are well explained and thorough. Although, there are some considerations that should be addressed:
Line 102: “solved” should be “dissolved”
Details regarding the gating strategy should be included either in section 2.5 or as a supplemental figure. For instance, is this data gated on all events or was debris gated out? Did the investigators do doublet discrimination?
Lines 161 and 165: Both Oxidative stress analysis and Cell membrane integrity assay should have more extensive methods. Were these kits purchased? Are there more extensive protocols published elsewhere that could enable other researchers to repeat these experiments?
Line 173: “72h” should have a space in between. Make sure to be consistent throughout the manuscript. Other examples are line 223 in Figure 1 legend and line 217.
Results:
The flow of the results section is streamline and the authors for the most part do not make any unsubstantiated conclusions based on their interpretation of the data. There were a few instances involving word choice that should be addressed; namely using the word proliferation when the data is reflective of viability which are two very different read-outs. Also, I am at a disagreement with the authors’ conclusions regarding dabrafenib treatment in figure 3 (see below). The main substance of this manuscript, namely monitoring changes in phenotypic switching protein expression, rests on the data shown in figure 7. Since this is the case, additional data should be provided. At the minimum, densitometry numbers should be suppled for all replicates used to generate the bar graphs.
Line 211: The authors state “little cell proliferation inhibition” but the read-out used is viability not proliferation per se. It is more accurate to state that there was little reduction in cell viability.
Figure 1: Following the comment from line 211, the concentration-response curves in figure 1 should have their axis changed from cell proliferation to cell viability. Additionally, a y-axis label should be added to the plots on figure 1c.
Line 217: double space between “celecoxib and” should be corrected.
Figure 2: I would recommend a positive control for future work especially for SK-Mel-28 which showed hardly any annexin V staining in figure 2. It would be beneficial for the reader to observe what a robust positive annexin V signal is from these cell lines.
Line 276: I disagree with this statement. SK-Mel-28 cells appear to be completely dead following 200 nM dabrafenib treatment whereas A375 look more confluent and less sensitive. These results contradict the data from figures 1 and 2. I would recommend reevaluating the authors interpretation of the data or their chosen representative image.
Line 280: The evidence provided from the light microscopy data is not a measure of cell proliferation. I recommend the authors revise this conclusion.
Line 287: Add a comma after “In both melanoma cells lines”
Line 288: Change “expecting” to “except” for better grammar
Line 293: I am unclear as to the what “faded” appearance means for SK-Mel-28. It would enhance the authors’ conclusions if they could elaborate on this observation and potentially add a reference for their conclusion.
Line 319: The manuscript would read better if the authors’ supplied a transitional sentence as to why they are interested in detected lipid oxidation particularly regarding the study of these drugs.
Line 376: The sentence, “At the centre of this phenomenon stays MITF expression that inversely correlates with AXL levels” should be restructured.
Line 452: Tyrosinase protein expression is used as a metric for melanogenesis. The conclusion sentence in this section is a too bold since the data is more suggestive and not definitive and it is recommended that the authors reflect that sentiment in their conclusions.
Discussion:
The discussion is extensive, and it outlines the major points the authors wish to convey. The authors demonstrate their great depth of knowledge in this area of research and provide appropriate references. Given there were some unexpected findings, the authors make great efforts to explain what might be going on while also acknowledging “how intricate melanoma biology actually is” (lines 569-570). Ultimately, enthusiasm for these results would be greater with additional studies. The investigators could for instance evaluate if other COX-2 inhibitors inhibit the Hippo pathway. Furthermore, authors could also conduct follow-up studies to verify if low dose celecoxib in fact induces a more invasive phenotype in these and other melanoma cell lines. More physiologic culture conditions, such as 3D spheroids may enhance their evaluation of Hippo pathway components due the nature of cell-cell contact influencing their activation status. With that said, the authors are for the most part very careful to not make unsubstantiated statements and acknowledge this is a preliminary study that warrants further investigation which is acceptable.
Line 477: Proliferation was not assessed in this study and the wording should be changed to reflect that.
Line 478: Add “and”; “melanogenesis, and correlated…”
Line 497: The authors should provide a reference supporting that ROS production might represent a sign of treatment resistance.
Line 529: Refence [20] is differently formatted from the other references and should be corrected.
Line 547: SK-Mel-28 proliferation rate was never measured only viability. The wording should be corrected to reflect that.
Reviewer 2 Report
Major Comments:
11) Authors must include error bars on the concentration response curves in Figure 1.
22) It is unclear what conclusions readers are supposed to draw from the microscopy images. The authors have only provided a few pictures with no quantification of any defined phenotypes. Authors must provide quantification/statistical analysis for these images.
33) In general, the writing is confusing and requires significant revisions to improve clarity.
44) The authors claim that there is an increase in apoptosis but performing an AnnexinV/PI experiment alone is not sufficient to make this claim. Additional experiments must be performed if the authors want to make this claim.
55) Authors must provide uncropped Western blots as supplemental figures.
66) The results from the Western blots are difficult to interpret for several reasons:
a. For the A375 cells the loading control (GAPDH) is uneven. It is not clear if the other proteins in this analysis are actually downregulated. Authors must perform additional experiments to clarify this point.
b. In many cases, the results of the Western blots are highly discordant between the two cell lines. The authors should attempt to explain this discordance, and the authors should also perform this experiment in additional melanoma cell lines.
77) The authors claim in the title that Celecoxib promotes “phenotype switching” in melanoma cells, but this claim is not supported by the data. Since the results are so discordant, you could just as easily make the conclusion that Celecoxib does not promote phenotype switching.
88) The authors should measure YAP1/TAZ nuclear localization and expression of YAP1/TAZ target genes by qRTPCR.
99) The authors must specify in the figure legends how many technical and biological replicates were performed for each experiment.
Minor Comments:
11) At multiple points in the manuscript, the authors claim that the cell lines are “BRAF positive”. Authors should be more specific about what that means. If the authors mean that the cells are positive for BRAF-V600E mutations, then that should be stated explicitly.
22) The authors did not comment on STR profiling or Mycoplasma testing.
33) The authors claim that the cell lines used in this study are “resistant to BRAFi” – I disagree with this point. First, other groups have derived BRAFi-resistant subclones from these cell lines, suggesting that the cell lines are not intrinsically fully resistant to BRAF inhibitors. Second, these cell lines are sensitive to nanomolar Dabrafenib concentrations whereas experimentally derived BRAFi-resistant cells are sensitive to micromolar concentrations of Dabrafenib. Finally, the authors did not profile any BRAF-WT/WT cell lines with Dabrafenib as a comparison. If the authors wish to make this point, then they must provide additional experimental evidence that these cells are actually resistant.
44) The authors claim that Celecoxib “significant increased the oxidative stress created by dabrafenib”. An alternative interpretation is that both Celecoxib and Dabrafenib increase oxidative stress, and it appears that the data better support this interpretation. The authors should provide a better discussion of whether there is truly synergy between the two compounds in this assay, and should provide additional analysis to demonstrate this point.
